# Routine Decontamination of Surfaces Relevant to Working Dogs: Neutralization of Superficial Coronavirus Contamination

**DOI:** 10.3390/ani12141823

**Published:** 2022-07-17

**Authors:** Sarah L. Grady, Natalie M. Sebeck, Mellisa Theodore, Karen L. Meidenbauer

**Affiliations:** Applied Physics Laboratory, Johns Hopkins University, Laurel, MD 20723, USA; sarah.grady@jhuapl.edu (S.L.G.); natalie.sebeck@jhuapl.edu (N.M.S.); mellisa.theodore@jhuapl.edu (M.T.)

**Keywords:** surface decontamination, virus neutralization, coronavirus, biological hazards, working dogs

## Abstract

**Simple Summary:**

The deployment of working dogs to environments containing potentially pathogenic biological agents necessitates a straightforward and fieldable decontamination protocol. This work describes the effectiveness of a wipe-based method using common veterinary cleaners on various surfaces contaminated with infectious virus. Surface characteristics, especially porosity, have a marked effect on the success of any decontamination protocol; however, in general, wiping with 70% isopropyl alcohol or 0.5% chlorhexidine proved to be most efficient at decontaminating surfaces common to the working dog community.

**Abstract:**

Given the increased deployment of working dogs to settings with pathogenic biological agents, a safe, effective, and logistically feasible surface decontamination protocol is essential to protect both the animals and their human handlers. Our group previously found that superficial contamination on surfaces relevant to the working dog community, including leashes and toys, could be significantly reduced using a standardized wiping protocol with various cleansing products. To expand upon this work, we analyzed the ability of this protocol to decontaminate surface-deposited bovine coronavirus, which was used as a BSL2 surrogate for SARS-CoV-2. Unsurprisingly, the physical characteristics of a given surface, including porosity and texture, had a significant effect on the ability to recover viable virus remaining on the surface post treatment. After correcting for these differences, however, wiping with 70% isopropyl alcohol (IPA) and 0.5% chlorhexidine performed best, reducing viral titers by >3 log on plastic bumper toys and nylon collars, and by >2 log on rubber toys and tennis balls. Leather leashes and Velcro proved more difficult to decontaminate, but both still showed significant loss of viral contamination following wiping with IPA or chlorhexidine. This work (i) validates the utility of a simple protocol for the neutralization of viruses on several surfaces, (ii) identifies materials that are more difficult to decontaminate, which should, thus, be considered for removal from field use, and (iii) highlights the need for further development of protocols testing porous or textured surfaces.

## 1. Introduction

Odor detection canines (ODCs) are frequently deployed to potentially hazardous environments to perform operational duties, which can range from the detection of explosives and narcotics to search and rescue efforts. Over the past several years, these animals have also been utilized in the medical field for detection of diabetes, specific cancers, and certain infections [1,2,3,4]. Unsurprisingly, with the emergence of the SARS-CoV-2 pandemic, ODCs are now even being considered a potential tool for identification of virus-infected individuals [5,6,7]. While deploying dogs to detect biological pathogens (i.e., “biosurveillance” missions) is of great interest, these operational environments carry with them an inherent risk for contamination, including deposition of virus particles on the ODC itself, its handler, or items being used during the completion of surveillance duties. This contamination presents a secondary infection risk to individuals that interact with these surfaces. With respect to the current pandemic, for example, it has been shown that infectious SARS-CoV-2 virions can be recovered hours to days following drying on various objects [8,9]. While fomite-based transmission has proven to be a less significant risk than person-to-person transmission for the existing strains of SARS-CoV-2 [10], it must still be considered, both to maximize safety under current conditions and to prepare for any future changes in viral transmissibility. There is also evidence that dogs can, in fact, act as a host for SARS-CoV-2, even if they appear to remain asymptomatic [11,12]. Together, this information emphasizes the need for effective decontamination protocols to be implemented on surfaces relevant to the ODC field.

Unsurprisingly, the majority of existing surface decontamination protocols focus on common household and clinical surfaces, including laminates, wood, stainless steel, and plastics, ignoring surfaces like leashes, dog toys, fur, or harnesses [13,14]. Even within the canine field itself, current decontamination protocols most often focus on single-incident scenarios and are frequently logistically burdensome (i.e., they are water-intensive, take >30 min, or require significant user-provided equipment) [15,16]. The creation of a fieldable and routine surface decontamination protocol for both the animal and the materials used during normal deployment, thus, remains a large gap that requires further investigation [17].

Previous work from our group showed that wiping various ODC-relevant surfaces with cloths dampened with select cleaning solutions provides an effective method for removal of gross superficial contamination, and the results discussed below expand upon this concept to provide pragmatic suggestions for treatments that result in viral neutralization [18]. Due to the ongoing SARS-CoV-2 pandemic, the team aimed to utilize a surrogate coronavirus that could be handled under BSL2 conditions, thus selecting the Mebus strain of bovine coronavirus. With this said, however, the strategy described herein could theoretically be adapted for use with any virus. The research was conducted using a robust, novel testing pipeline that could easily be modified in the future to (i) evaluate the ability of any solution to inactivate a wide range of viruses spiked onto surfaces, and (ii) identify scenarios where the use of alternative materials should be considered because currently used surfaces prove difficult to clean (or cleaning effectiveness is difficult to accurately measure).

Given the significant interest in applying ODCs toward detection of human infections, it is likely that they will continue to be deployed to environments potentially contaminated with biological threats. The safety of the animal and its handler must remain a priority for the agencies utilizing these critical detection teams. By balancing decontamination potential and logistical burden, this work can help guide the field through the end of the current pandemic while informing best practices for future emerging threats.

## 2. Materials and Methods

### 2.1. Virus Production and Enumeration:

Bovine coronavirus (bCoV) strain Mebus (NR-445) was obtained from the Biodefense and Emerging Infections (BEI) Research Resources Repository and propagated in HRT-18G cells (ATCC CRL-11663). Cells were maintained in growth media containing Eagle’s modified essential medium (EMEM) supplemented with fetal bovine serum (FBS) at concentrations between 2% and 10% and incubated at 5% CO_2_ and 37 °C. Virus samples were enumerated using the tissue culture infectious dose 50% (TCID_50_) assay [19].

### 2.2. Tested Surfaces

Tested surfaces were selected to allow for direct comparison with the results of the gross contaminant removal study performed by the authors previously [18]. Coyote pelts (Paulette Fur Co., Malvern, OH, USA) served as a surrogate for domestic dog fur to avoid viral contamination of living animals and to allow for sterilization prior to inoculation. Velcro harnesses, nylon collars, and leather leashes were components of the TSA Handler kits (Ray Allen Manufacturing, Colorado Springs, CO, USA, RAM-K-TSA-2H). Tennis balls (Quiet Glide, Hoover, AL, USA, pre-cut, gray, T34GRY20), rubber Kong toys (KONG, Golden, CO, USA, 53352), and dummy bumpers (SportDOG, Knoxville, TN, USA SAC00-11672) were selected to represent common dog reward items.

### 2.3. Neutralization Solutions and Wipes

Similar to test surfaces, candidate neutralization solutions were chosen on the basis of successful testing in the previous removal study. Safety, accessibility, and antimicrobial/antiviral activity were also considered during the down-selection process. All compounds were tested at concentrations deemed safe to animal and handler, and which were not reported in the literature to cause nose blindness in dogs. The following neutralization solutions were utilized: deionized Milli-Q water, Pampers brand sensitive baby wipes, 70% isopropyl alcohol (Equate, Houston, TX, USA, FG003032), and 2% chlorhexidine gluconate (Durvet, Blue Springs, MO, USA, 30798-624-35, diluted 1:4 in deionized water for a final concentration of 0.5%). Disposable soft-spun dry fabric wipes (Medline, Northfield, IL, USA, 1013) were used for all experiments.

### 2.4. Cytotoxicity of Neutralization Solutions

Serial dilutions of each neutralization solution were prepared in growth medium and applied to HRT-18G cells. Cells were incubated for 6 days at 37 °C and 5% CO_2_ to mimic the length of time necessary to complete the TCID_50_ titering assay. After the incubation period, cell viability was determined using the WST-8/Cell Counting Kit 8 (Abcam, Boston, MA, USA, 228554) following the manufacturer’s recommendations. Briefly, the WST-8 solution was diluted 1:10 in EMEM without phenol red, and 100 μL of the resulting mixture was added to each well. Plates were incubated in the dark for 2 h at 37 °C. Absorbance was measured at 460 nm using a TECAN Spark spectrophotometer.

### 2.5. Dye Recovery Experiments

A 10 mg/mL solution of trypan blue in growth media was used as a proxy for viral stock solutions to allow for preliminary high-throughput testing during down-selection of wiping or swabbing methods. Five closely spaced 10 μL droplets were applied to 1” × 1” coupons of each surface and permitted to dry for 60 min at ambient temperature and humidity inside a biosafety cabinet (BSC). Coupons were processed following one of two pipelines. In the “vortex” pipeline, coupons were placed directly into a 50 mL conical vial containing 5 mL of growth medium and vortexed at maximum speed for 30 s. In the “swab” pipeline, a clean cotton swab was dipped in 0.5 mL of growth medium and used to wipe a coupon three times in the vertical direction and three times in the horizontal direction. The liquid in the swab was expressed back into the conical vial. All solutions were serially diluted, and absorbance was measured at 600 nm using a TECAN spectrophotometer.

### 2.6. Virus Neutralization on Surface Coupons

All surfaces were cut into 1” × 1” coupons and sterilized by submersion in a 70% ethanol bath for 15 min followed by overnight drying in a BSC. Five closely spaced 10 μL droplets of viral stock were deposited on each surface and allowed to dry in a BSC for 60 min at ambient temperature and humidity. The wiping protocol followed the same approach used in the group’s previous removal study, and all manipulations were performed under sterile conditions within a BSC [18]. Briefly, 500 mL of each neutralization solution was decanted into a clean container, dry wipes were fully submerged in each solution, and the resulting wet wipes were gently squeezed to remove dripping liquid. The liquid contained in baby wipes was expelled into a container and transferred on to the same wipe material as used for all other solutions. All experiments were performed by the same researcher to minimize any differences in pressure or technique between replicates. Three wiping motions were applied to each coupon using the same side of a cloth in vertical direction. Each cloth was then folded in on itself, and the coupon was wiped three additional times in the horizontal direction using the clean side of the cloth. Gloves were replaced after each surface was tested.

Wiped coupons were allowed a brief drying time of 5 min, and then placed into a 50 mL conical vial containing 5 mL of EMEM + 2% FBS. The conical was vortexed at high speed for 30 s, and the resulting supernatant was frozen at −80 °C prior to viral enumeration.

### 2.7. Virus Neutralization Following Direct Contact

A single 10 μL droplet of virus stock was deposited in an Eppendorf tube, and 50 μL of neutralization solution was placed directly on top of the droplet. This mixture was allowed to incubate for 5 min at ambient temperature before resuspension in 1 mL of growth medium. The resulting solution was stored at −80 °C prior to viral enumeration.

### 2.8. Statistical Analysis

The *p*-values were determined using a two-tailed Student’s *t*-test assuming unequal variance.

## 3. Results

### 3.1. Droplet Drying Rates and Recovery Strategy

To mimic a potential viral contamination event, 10 μL droplets of growth medium were spiked and allowed to dry on several surfaces relevant to the working dog community. Tested surfaces included coyote pelts, nylon collars, Velcro harnesses, leather leashes, rubber Kong toys, plastic dummy bumper toys, and tennis balls. While differences in physical characteristics between surfaces caused drying rates to differ, all coupons appeared visually dry by 60 min (Figure 1). Constituents in the growth medium did tend to leave an apparent residue on surfaces that did not disappear, even following >24 h of drying time. As such, the 60 min drying period was utilized for all subsequent testing.

To determine the optimal protocol for recovering viable virions from each surface, a solution of trypan blue was used as a proxy for viral stocks, as its concentration could be rapidly quantified by spectrophotometric measurements. Vortexing proved to be the more efficient recovery method across all surfaces with the exception of the leather leash, which showed poor recovery regardless of treatment (Figure 2). On the basis of these results, the vortexing protocol was used for all neutralization trials involving infectious virus. 

### 3.2. Cytotoxicity of Neutralization Solutions

To effectively test the ability of each candidate solution to neutralize infectious viruses, an appropriate virus strain needed to be selected. With the SARS-CoV-2 pandemic altering practices across the globe, the selection of a BSL2 coronavirus seemed most relevant. Bovine coronavirus (bCoV) easily replicates to high titers in mammalian tissue culture and shares many of the same physical characteristics of SARS-CoV-2, including approximate particle size, the presence of a lipid envelope, a positive-sense RNA genome, receptor binding affinities, and efficacy of inactivation using conventional disinfectants [20,21,22]. Additionally, the genetic similarity between bovine and human coronaviruses is thought to be sufficient to allow for recombination/spillover, and bCoV has been utilized as a SARS-CoV-2 surrogate in other studies [23,24]. Considering these characteristics, bCoV was chosen as the virus used for all downstream experiments. 

The TCID_50_ readout assay used to titer bCoV requires the measurement of cytopathic effects (CPEs) induced during the viral replication cycle. To differentiate between cell death due to the virus and cytotoxicity induced by remnants of the neutralization solutions transferred onto cells following the vortexing protocol, cell viability assays were performed using twofold serial dilutions of each candidate solution. While no cytotoxicity was observed with baby wipe liquid at any dilution, the neat 0.5% chlorhexidine solution caused significant cell death, and the 70% IPA solution showed observable, although not statistically significant, effects on cell death measurements up to the 1:8 dilution factor (Figure 3). For this reason, and to facilitate subsequent titration calculations, IPA- and chlorhexidine-treated samples were diluted by tenfold prior to performing TCID_50_ assays. The resulting limit-of-detection values for the TCID_50_ assay were 100 infectious bCoV particles for IPA/chlorhexidine samples and 10 infectious bCoV particles for samples treated with water or baby wipe liquid.

### 3.3. Virus Neutralization by Wiping

Both the droplet drying process itself and the physical characteristics of each tested surface were predicted to have an effect on the efficiency of recovery of infectious bCoV particles across trials. To determine how drying affected viral titers, droplets were dried inside of an empty conical tube and recovered through the direct addition and vortexing with growth media. This resulted in the loss of about 60% of the theoretical max yield, from 1.58 × 10^6^ TCID_50_/mL for the virus stock solution to 6.69 × 10^5^ TCID_50_/mL. This loss of titer has been similarly observed following drying of other enveloped viruses [25]. To determine the additive effect of drying and vortexing a coupon with minimal porosity, droplets were dried and recovered from the sterile lid of a tissue culture plate. Using this surface, virus titers were reduced by approximately 65%, from 1.58 × 10^6^ TCID_50_/mL to 5.49 × 10^5^ TCID_50_/mL (Figure 4A). This latter value was set as the new “max theoretical yield” for the neutralization experiments.

Control trials with each surface showed recovery efficiencies ranging from approximately 1% for leather to 34% for nylon (Figure 4B). In all cases, recovery was significantly decreased when compared to tissue culture plates. Virus was not recovered from coyote fur pelts; hence, no further testing could be performed on this surface. The variability in recovery efficiencies likely represents the sum effects of (i) the porosity/hydrophobicity of the surface, and (ii) the potential antiviral activity of a component of the surface itself. The independent efficiency of each neutralization solution to decontaminate each surface was based on the further decrease in yield from these baseline control values.

Across all surfaces, the 70% IPA and 0.5% chlorhexidine solutions proved most effective at neutralizing bCoV (Table 1). IPA wiping reduced viral titers by greater than 2 log (>99%) on three surfaces (tennis ball, Kong, Velcro) and greater than 3 log (>99.9%) on two surfaces (bumper and nylon). It performed least effectively on leather, but still reduced titers by greater than 1 log (>90%) on this surface. In the case of the bumper, tennis ball, nylon, and Velcro samples, the actual reduction in titers could have been greater than the measured value, but the limit of detection of the assay would not allow for absolute quantification. 

The chlorhexidine solution reduced bCoV titers by greater than 2 log (>99%) on the Kong and tennis ball, by greater than 3 log (>99.9%) on the bumper and nylon, by approximately 95% on Velcro, and by approximately 70% on leather. Water and baby wipe liquid neutralized virus on the Kong, bumper, and nylon surfaces relatively well (greater than 2 log or more than 99% reduction), but performed less effectively on the tennis ball (~75% reduction), Velcro (~55% reduction), and leather (no measurable reduction) (Table 1). Taken together, these results suggest that (i) wiping with IPA or chlorhexidine solutions provides an efficient way to neutralize bCoV contamination on a variety of operationally relevant surfaces, (ii) Kongs and bumpers are the most easily decontaminated surfaces, and (iii) leather proves to be the most difficult of the tested surfaces to decontaminate.

### 3.4. Virus Neutralization by Direct Contact

Under operational conditions, viable virus particles could remain on wipes after their use, posing a continued hazard to handlers. To determine whether a wipe impregnated with a neutralization solution would render virus nonviable within 5 min, a direct contact experiment was performed. Here, a 50 μL droplet of neutralization solution was applied directly on top of an undried 10 μL droplet of virus stock for 5 min. This mixture was then diluted in growth medium and titrated. While direct contact incubation of the virus with negative control and baby wipe solutions had no effect on viral viability, both the 0.5% chlorhexidine and 70% IPA solutions reduced viral titer by at least 2 log or 99% (Figure 5).

## 4. Discussion

The effective neutralization of viruses on surfaces relevant to the working dog community is of high concern, as these animals and their handlers are increasingly present in potentially contaminated environments [26,27,28,29]. This study aimed to determine the extent to which wiping with four specific solutions reduced the recovery of viable bovine coronavirus particles from various surfaces relevant to the ODC community. Previous work from our group found that the act of wiping many surfaces with any moistened cloth is an effective strategy for removing gross superficial contamination [18]. This work supports that conclusion, as water-wetted cloths neutralized close to 3 log (99.9%) of viable virus from three of six tested surfaces (rubber Kong toy, plastic dummy bumper, and nylon harness). However, 70% IPA and 0.5% chlorhexidine solutions were more effective for virus neutralization across a wide variety of surfaces, including the three porous and/or more high textured surfaces that were tested (tennis balls, Velcro harnesses, and leather). Operationally, considerations also need to be paid to whether, and for how long, viable virus remains on any wipe used by handlers. Direct contact experiments in the present study showed that 70% IPA and 0.5% chlorhexidine are the best options, as a 5 min incubation reduced virus yield to below the limit of detection of the assay. Taken together, these results suggest that, while any kind of wiping is useful, disinfectant solutions are the more effective option when aiming to (i) decrease viable virus on possible fomites, and (ii) reduce the risk of secondary infection due to handling contaminated surfaces.

This work also emphasizes the importance of material selection when creating and purchasing canine equipment. The leashes currently utilized by the TSA, for example, have leather components that proved difficult to disinfect using the strategies deployed by this study. Shifting from leather to an alternate nonporous material could provide a simple solution to minimize contamination and unnecessary viral exposures. Additionally, this study showed that a cost–benefit analysis should be performed on cleaning versus discarding canine equipment. While some equipment is expensive and should be repeatedly disinfected to avoid an untenable budget impact, it may prove easier to throw away toys like tennis balls after each trip to the field.

Finally, this body of work further emphasizes the need for continued protocol development in the field of porous surface decontamination. While viral binding to porous surfaces may reduce the risk of fomite-based transmission [30,31,32,33,34], it is unlikely that this risk is reduced to zero. The authors attempted multiple strategies to recover viable virus from fur, including increasing droplet size, trimming the length of the fur on the pelt, changing the fur material (coyote, raccoon, and synthetic), and increasing the amount of applied virus. No viable virus was recovered under any tested condition. While this issue has been observed, to variable degrees, in multiple other reports [35,36,37,38,39], in order to have high confidence in any neutralization/decontamination protocol to be used on live animals, further work is necessary to optimize recovery from this difficult surface. It is also important to note that fur on live animals is likely maintained at a higher temperature than the ambient air, and that this difference could and should be implemented in future laboratory testing.

## 5. Conclusions

This study confirmed that a simple, wipe-based protocol using 70% IPA or 0.5% chlorhexidine is likely sufficient for routine viral decontamination of most surfaces used in the ODC community. This strategy is logistically feasible, and the pipeline used to generate these results is amenable and adaptable to testing against other surfaces, contaminants, and decontamination strategies. Continued improvements in strategies for evaluating porous or textured surfaces remains necessary; however, with advancements in this arena, pragmatic decisions about surfaces selected for use in different environments can be made.

## Figures and Tables

**Figure 1 animals-12-01823-f001:**
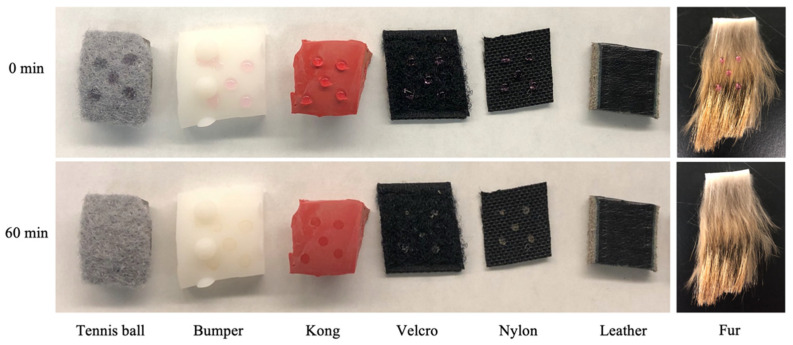
Droplet drying rates. Images show the effect of 0 and 60 min drying times on 10 μL droplets of growth medium deposited on various surfaces. Drying occurred at ambient temperature and humidity inside a biosafety cabinet.

**Figure 2 animals-12-01823-f002:**
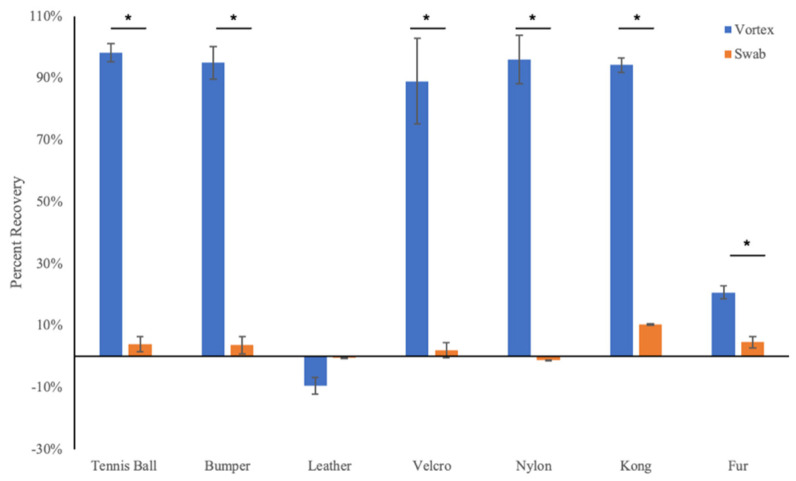
Vortexing using growth medium is more efficient than wet swabbing for recovery of trypan blue dye dried on various surfaces. Values are expressed relative to the absorbance of growth medium spiked directly with trypan blue. *N* = 3; * *p* < 0.05.

**Figure 3 animals-12-01823-f003:**
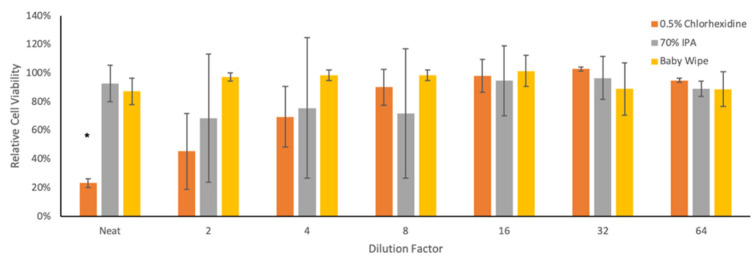
Cellular cytotoxicity of neutralization solutions. Values are expressed relative to samples exposed to the same dilution of water in growth medium. *N* = 3; * cell viability significantly decreased relative to corresponding water control (*p* < 0.05).

**Figure 4 animals-12-01823-f004:**
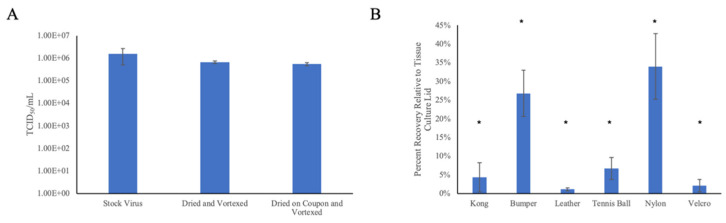
The effect of drying and surface characteristics on recovery of viable virus. (**A**) Recovery of viable virus from droplets placed inside conical tube or on a tissue culture plate lid. All samples were dried for 1 h prior to and rehydration with growth medium. *N* = 2. (**B**) Percentage recovery of viable virus on test surfaces relative to recovery from lid of tissue culture plate. *N* = 3; * recovery was significantly decreased when compared to this control (*p* < 0.05).

**Figure 5 animals-12-01823-f005:**
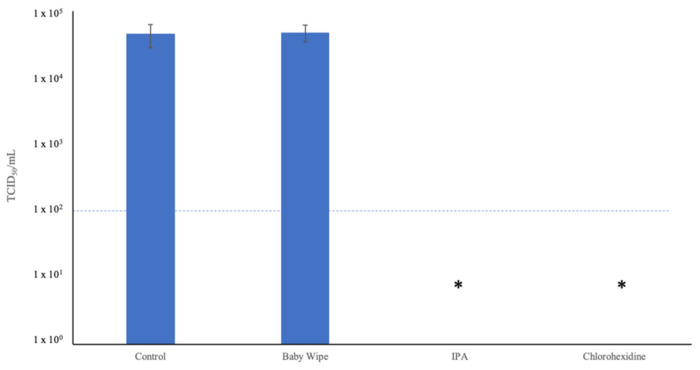
Effect of direct contact of neutralization solutions on dried virus droplets. *N* = 3; * titer fell below the limit of detection of the assay, meaning absolute reduction in virus yield could not be determined.

**Table 1 animals-12-01823-t001:** Effect of wiping with neutralization solutions on dried virus droplets deposited on test surfaces. *N* = 3; * titer fell below limit of detection of the assay, meaning absolute reduction in virus yield could be greater than listed.

Surface	Percent Reduction in Viable Virus			
	Water	Baby Wipe	0.5% Chlorhexidine	70% IPA
Kong	>99.9 *	>99.9 *	>99.0 *	99.0
Bumper	99.8	99.6	>99.9 *	>99.9 *
Tennis Ball	79.2	72.8	>99.0 *	>99.0 *
Leather	0	0	70.4	95.0
Nylon	99.7	99.9	>99.9 *	>99.9 *
Velcro	59.3	52.0	95.6	>99.0 *

## Data Availability

Not applicable.

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
