# Peer review of "Routine Decontamination of Surfaces Relevant to Working Dogs: Neutralization of Superficial Coronavirus Contamination"

_animals, 2022, doi:10.3390/ani12141823_

Round 1

Reviewer 1 Report

Review of “Routine decontamination of surfaces relevant to working canines: neutralization of superficial coronavirus contamination”

The authors present their results from a study comparing various methods of surface decontamination as pertaining to situations commonly encountered by working dogs.

As a general comment, there is no reason to refer to dogs as “canines” in this context; the term “dog” is less vague and more specific to what the authors are describing and thus preferable, please replace throughout.

As another general comment, it is not appropriate to hide the fact that the authors used a bovine coronavirus in the methods section. In the context of this paper, “coronavirus” is automatically assumed to refer to SARS-CoV-2, and any other coronavirus needs to be specifically identified in a prominent place.

Abstract

Viruses are not alive, please use terms such as “active” or “infectious” instead of “live”.

You are neutralizing the virus by decontaminating/disinfecting the material. You are not neutralizing the material itself.

Keywords: include “surface disinfection”, “virus neutralization” and/or other more specific keywords.

Introduction

This section requires a paragraph expanding on what we know about SARS-CoV-2 persistence on surfaces and its importance as an infectious pathway for humans. The authors should also expand on their assertion that the virus represents a hazard to the dogs themselves as opposed to their handlers and specify how exactly that is the case based on the literature.

Line 44, define what you mean by “biosurveillance”.

M&M

As above, it is inappropriate to refer to the bovine coronavirus the authors used simply as “coronavirus” in the title and the rest of the paper. That term currently is largely used to refer to SARS-CoV-2, therefore any other coronavirus requires specific identification.

Line 118, specify what measures were taken to make sure these samples were treated aseptically following sterilization.

Results

This section does not appear to have any significance calculations or P values, please provide appropriate statistical tests comparing the outcomes between different cleaning solutions and surfaces.

Line 167 and following, the rationale for using bCoV would best be moved to the introduction.

As a comment on the results from fur, I would point out that fur on a live dog is probably at a higher temperature than what the authors used due to dissipating body heat.

Discussion

Expand a bit more on how exactly your virus behaves similarly to SARS-CoV-2 based on what the literature shows.

Reviewer 2 Report

This manuscript described about using different neutralization solution to remove coronaviruses from tested surface. In general, the experiments are well design and thoughtful. However, the major concern is this manuscript is highly similar with previous publication by author research team (Routine Decontamination of Working Canines: A Study on the Removal of Superficial Gross Contamination. Health Secur 2021, doi:10.1089/hs.2021.0070),  in which Glo Germ MIST (Glo Germ Company, item RFMST) was replace with Bovine coronavirus (bCoV) strain Mebus (NR-445) in this manuscript.

Both experiments indicated that 0.5% chlorhexidine solution and 70% isopropyl alcohol were the most effective to disinfect, leather components is the most difficult to disinfect. The result and findings are very similar in these 2 manuscripts.

Other comments are:

Line 46~ what is the meaning of logistically burdensome?

Line 80~ suggested to input the figures of the items use, most people will not understand KONGS

Figure 1 ~ can’t really appreciate the droplets on Leather (at 0 min)

Figure 2 ~ for Y-axis, the meaning of 110%? And the meaning of -10%? How to measure -10%?

Figure 3 ~ the meaning of 120% cell viability?

Line 280 -281 ~ the statement “While some equipment is expensive and should be repeatedly disinfected to avoid untenable budget impact, it may prove easier to throw away toys like tennis balls after each trip to the field.” Is similar with statement in journal (Routine Decontamination of Working Canines: A Study on the Removal of Superficial Gross Contamination. Health Secur 2021, doi:10.1089/hs.2021.0070

Round 2

Reviewer 1 Report

Thank you for addressing my concerns, this is acceptable for publication.

Reviewer 2 Report

I satisfied with the amendment on the manuscript.